# Ultrasound-Based Smart Corrosion Monitoring System for Offshore Wind Turbines

Upeksha Chathurani Thibbotuwa [1,*], Ainhoa Cortés [1,2,*] and Andoni Irizar [1,2]

1   CEIT-Basque Research and Technology Alliance (BRTA), Manuel Lardizabal 15, 20018 San Sebastian, Spain; airizar@ceit.es

2   Department of Electronics & Communications, Universidad de Navarra, Tecnun, Manuel Lardizabal 13, 20018 San Sebastian, Spain

*   Correspondence: uthibbotuwa@ceit.es (U.C.T.); acortes@ceit.es (A.C.); Tel.: +34-943212800

**Abstract:** The ultrasound technique is a well-known non-destructive and efficient testing method for on-line corrosion monitoring. Wall thickness loss rate is the major parameter that defines the corrosion process in this approach. This paper presents a smart corrosion monitoring system for offshore wind turbines based on the ultrasound pulse-echo technique. The solution is first developed as an ultrasound testbed with the aim of upgrading it into a low-cost and low-power miniaturized system to be deployed inside offshore wind turbines. This paper discusses different important stages of the presented monitoring system as design methodology, the precision of the measurements, and system performance verification. The obtained results during the testing of a variety of samples show meaningful information about the thickness loss due to corrosion. Furthermore, the developed system allows us to measure the Time-of-Flight (ToF) with high precision on steel samples of different thicknesses and on coated steel samples using the offshore standard coating NORSOK 7A.

**Keywords:** corrosion monitoring; FPGA; offshore wind turbines; ultrasound; thickness loss

## 1. Introduction

Wind energy plays a significant role as a renewable energy source, urging the need for cost-effective renewable energy sources. Compared to onshore wind energy, offshore wind turbines benefit from two main advantages: higher mean wind speeds (modest increases in wind speed can result in doubling the generated power) and steadier wind supply, which makes power generation more reliable. These two factors combined have a dramatic effect on the Return on Investment (RoI) of the farm [1].

In the last decade, we have witnessed a steady increase in the total installed capacity of offshore wind farms. By the end of 2019, 6.1 GW capacity has been newly installed. The total capacity installed by 2019, was 29.1 GW, which is a 10% increase with respect to 2018 [2]. Europe (UK, Germany, Denmark, Belgium) is leading by contributing 75% of total global offshore wind installation, as of the end of 2019, and Asian offshore wind energy production keeps growing significantly, lead by China and followed by Taiwan, Vietnam, Japan, and South Korea. Due to COVID-19, the overall expected wind capacity in 2020 was not met because of the challenges in project construction, execution activities, and investments. However, the Global Wind Energy Council (GWEC) has predicted that offshore wind energy will have a 20% of contribution to the global wind installation by the year 2025.

Offshore structures are installed in relatively deep water and exposed to the harsh marine environment. This leads to an increase of the cost of construction and deployment compared to onshore structures [3] and higher operation and Maintenance (O&M) costs of installations compared to land or coastal-based structures due to the limitations in accessibility, manpower, special equipment requirements, etc. [4]. The added costs in the construction and deployment of offshore turbines were mitigated at the beginning as the

turbines were installed mainly in coastal sites with shallow waters. In recent years, there has been a constant increase in the water depth and distance to shore for the offshore wind farms [5], as farms are located further away from shore (60 km) and as depth water reaches 200 m [6]. This leads to higher Capital Expenditure (CAPEX) and O&M costs that reduce the benefits of the farm [7]. Several foundations for wind turbines have been developed trying to reduce the construction and installation costs. Another key to the profitability of the wind farm is to improve the useful life of the wind turbines by maintaining the operational conditions of the towers to minimize the breakdowns or downtime of the turbines. Therefore, improving O&M models and tools and, at the same time, reducing costs have become key to maintain the profitability of offshore wind farms. Numerous research works providing different approaches that support directly or indirectly to improve O&M process of offshore wind turbines/farms have been discussed in Reference [8–12].

### 1.1. Structural Health Monitoring of Wind Turbines: Corrosion Monitoring

Structural Health Monitoring Systems (SHMSs) for Offshore Wind Turbines (OWTs) are used to detect damage in blades [13,14], tower, and support structure. Since the tower and foundation are two key elements of OWT that support the structural integrity, they must operate as much time as possible. Once the turbine is installed, it should sustain with associated loads, and their partial failure would carry catastrophic consequences. Not much progress has been made in developing robust applications regarding SHMs for operating WTs [15].

Corrosion is one of the main root causes that can produce offshore structural failure. Corrosion monitoring is the process of obtaining very frequent corrosion measures to evaluate the progress of corrosion within a specific environment. Corrosion inspections are done to evaluate the material condition at any given time based on a pre-scheduled routine or the available risks. Usually, inspections are performed much less frequently than corrosion monitoring. Frequent corrosion measures are always advantageous of early detecting the risks, repairing conditions, and, consequently, reducing the operational and maintenance cost associated with corrosion. The rust formation due to corrosion depends on the environment and the type of material, as well. Carbon steels corrode mostly by general corrosion (uniform corrosion), but localized types of corrosion can also take place, e.g., pitting, which usually are not considered in conventional corrosion analysis [16]. Different reasons can be identified behind the growth of corrosion process in an offshore structure, such as temperature, salinity, pH, and coating damage due to tidal fluctuations, dissolved oxygen, variable cyclic load due to wave and wind impact, etc. [17].

Even for a well-designed structure, with a long time of exposure to the harsh marine environment, corrosion causes the degradation of the metal leading to create corrosion fatigue cracks and, consequently, structural failure. This scenario increases the O&M cost because of the possibility of more frequent maintenance, repair activities, and any replacement if needed [18]. Therefore, a successful corrosion monitoring approach can potentially decrease the ongoing O&M cost of wind turbines directed at reducing economic losses and environmental damages. However, corrosion monitoring is one of the major challenges that an SHM system developed for an OWT faces, mainly due to the accessibility difficulties and the area the corrosion monitoring system must cover, which is very large even when one considers that some zones are more affected than others, e.g., the splash zone. Thinking about an automated solution, ideally, the corrosion sensor should work unattended 24/7 for several months or even years. This imposes very hard specifications that the monitoring system must meet. Therefore, prior to the selection of an appropriate corrosion monitoring technique and sensors, the zone of interest (splash zone, atmospheric zone, submerged zone) of the wind tower and the respective environmental conditions must be considered separately [19].

### 1.2. Corrosion Inspection and Monitoring Techniques

There are different types of corrosion inspection and monitoring techniques to find out the corrosion condition of metals, as is discussed and reviewed in Reference [20,21]. Figure 1 shows a detailed classification of these techniques based on the corrosion detection method and sensing parameters.

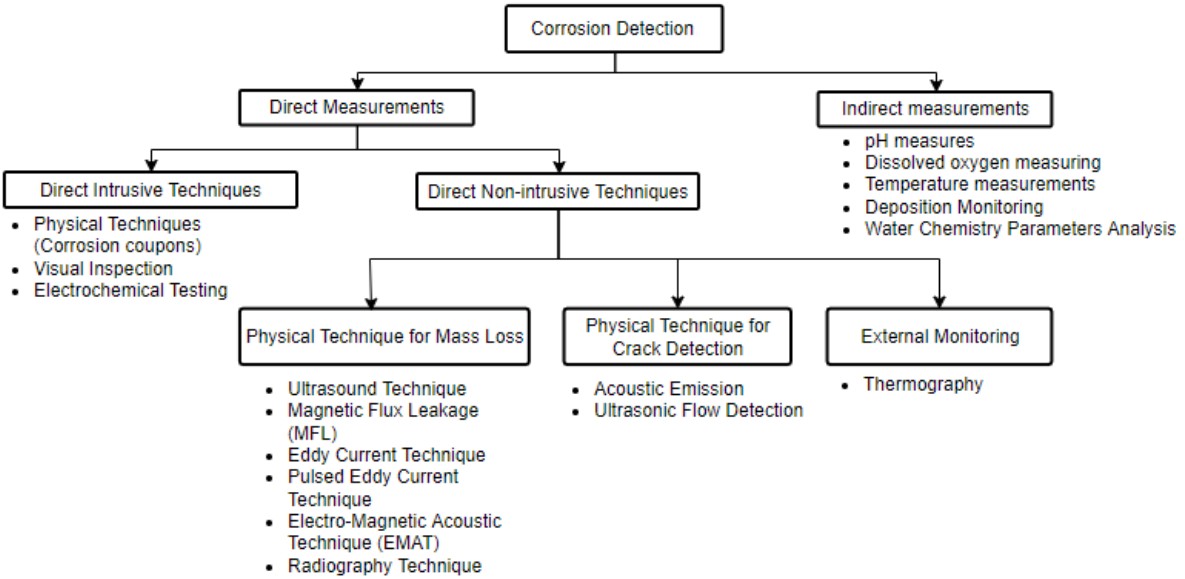

**Figure 1.** Corrosion detection methods.

In general, corrosion detection measures can be direct measuring metal loss due to corrosion or corrosion rate or indirect measuring any parameter that may be a cause or consequence of the metal loss or corrosion. Some of these techniques need direct contact/exposure to the same corrosive fluids or to access the internal environment where the corrosion process takes place. They are known as intrusive techniques and can alter the corrosion process and create disturbances to the operation during installation and re-installation of measuring probes or inspection processes. Therefore, it is beneficial if corrosion testing can be performed from the outside of the test object. This is possible with non-intrusive techniques through detecting physical and geometry changes due to corrosion (mass loss, crack, or surface discontinuities), and there is no need for any direct contact to the same corrosive fluids or access to the internal environment. On the other hand, it is very important to perform corrosion detection activities in the field applications without destroying the engineering properties of the material and without affecting its long-term future performance during the inspection process. This kind of evaluation can be achieved by using Non-Destructive Testing (NDT) methods for material testing [22]. Among the techniques listed in Figure 1, if this condition is satisfied, they are known as an NDT method. Therefore, we can propose that the most appropriate corrosion detection technique for offshore wind turbines shall be both non-intrusive and non-destructive.

The main objective of this research work is to develop a smart ultrasound solution for corrosion monitoring in offshore wind turbines. Due to the harsh and distant environment they have been installed in, it is a challenge to find the most suitable corrosion detection technique. Based on the expected solution, we have identified the following design specifications to be achieved and feasible with the selected corrosion detection method.

- SHM requires a reliable and permanent acquisition of monitoring data with minimum interruptions.
- Corrosion detection technique shall be non-intrusive and non-destructive.
- The monitoring system is able to detect uniform corrosion.

- The solution is able to detect corrosion in both splash zone and atmospheric zone of the wind turbine tower.
- Data acquisition must run for several years in order to obtain meaningful results about the tower's health.
- Robust wireless communications are needed to rely on the sensor data sent to the tower's central computer in a difficult environment for electromagnetic signals.
- Corrosion detection components must have the capability of integrating with Unmanned Aerial Vehicles (UAVs) as they offer significant potential benefits to the monitoring of large-scale facilities, e.g., atmospheric zone.
- Tower locations which are not easily reached by a mobile platform must be covered by battery powered sensor nodes fixely installed in those areas, e.g., splash zone.

The paper is structured as follows. In Section 2, a review of different ultrasound solutions for corrosion monitoring is introduced. Section 3 describes the concept of the presented approach for corrosion monitoring in offshore wind turbines. Section 4 provides the description of the ultrasound testbed and the design methodology followed to miniaturize the solution. In Section 5, the samples preparation and the experimental setup are presented. Section 6 shows the most relevant results we have obtained through our experiments. Finally, the conclusions are summarized in Section 7.

## 2. Related Work

In ultrasound techniques, a high-frequency ultrasound wave will travel along with a material that shows elastic properties. The frequency range of ultrasonic sound waves used for NDT of materials is generally 1 MHz to 15 MHz [23]. The velocity of the ultrasound wave propagation through a material is a function of the elastic modulus and density of the material [22,24]. Importantly, ultrasound waves are distinctly reflected at the boundaries when the acoustic impedance change due to a different material or medium properties. Based on that, the thickness loss due to corrosion will be given as a function of propagating velocity, frequency, and energy components [25]. The setup can be designed based on two common techniques [23,26]:

1. Pulse echo method: A short period of ultrasonic energy is introduced into the test piece with regular pause, and the reflected part of the ultrasonic wave is utilized for the analysis. As the sender and receiver probes use the same transducers, either one transducer or two transducers placed on the same side can be used.
2. Through transmission method: The transmitted part of the ultrasound wave is used for the analysis. Separate transducers placed on opposite sides are used as the sender and receiver probes.

It can be said that the best technique for corrosion monitoring would be the pulse-echo method as it gives the depth/location of the defect in comparison to the through transmission method.

Thus, current sensing methods using the ultrasound technique give valuable information about various forms of degradation that occur in all sorts of materials (cracks, deformations, thinning, corrosion, etc.), but, in general, they are classified as inspection devices, rather than monitoring devices. The inspection devices are usually "once-off" measurements that cover a given period of time (according to maintenance schedules), while the monitoring devices involve very frequent measurements that must detect small fluctuations in the advancement of corrosion in order to be used as input for a general assessment of corrosion [27]. Several previous works have been reported to assess corrosion in pipelines using ultrasounds [28].

Although not focused on corrosion detection, Ref. [13] proposes a novel walking robot-based system for NDT in wind turbines. Moreover, commercial systems based on ultrasounds exist in the market to monitor large structures. For example, Eddyfi Technologies has developed the Scorpion2 product [29] for ultrasonic tank shell inspections. To the best of our knowledge, it monitors wall thickness during the inspection using a dry-coupled, remote-access ultrasonic crawler that brings major efficiency and data

improvements to tank shell inspections. It scans the tank wall to get an image of the wall status with the aim of estimating the wall thickness. Among other NDT techniques, the walking robot offers ultrasonic testing, but it is aimed at identifying cracks, delamination, etc., in composites (e.g., in blades). Furthermore, this walking robot has already been used in onshore wind turbines.

Ultrasound is a technique with a lot of potentials to deploy a Wireless Sensor Network (WSN). Nodes can be housed in small devices with communication capabilities, drawing very little power from a battery and offering very good corrosion detection capabilities. However, this is not enough when referred to large structures (e.g., offshore WT) where corrosion may develop randomly at a given point. The continuous technological improvements of all kinds of unmanned vehicles (terrestrial, maritime, and aerial) during the last decade make this technology very attractive for many industries that require frequent and costly maintenance operations in areas where access is difficult. Ref. [30] proposes an autonomous ultrasonic inspection using UAVs. This research is very relevant since it demonstrates the influence of the accuracy of the measurements positioning and of the electrical noise produced by control electronics and the propeller motors into the thickness estimation. These uncertainties cause ultrasonic signal coupling issues. One of the main advantages of permanently installed ultrasound sensors is that they achieve a better repeatability since the coupling and probe-positioning errors are removed. Hence, in the case of mobile solutions, there are challenges to solve regarding probe alignment, more robust control strategies, enhanced positioning the accuracy, and novel ultrasonic angular coupling capabilities. The paper is focused on the improvement of the UAV design, and they do not show the accuracy obtained in the thickness estimation. Additionally, in this paper, they claim to detect the loss of thickness over non-coated aluminium samples (11 mm of thickness). It is well known that coating reduces accuracy of wall thickness estimations [31], and wind turbine walls are usually protected with standard coatings of different thicknesses and formulations. Therefore, coating affects the ultrasound signal response, and it is another key challenge to be solved, not only for ultrasound mobile solutions but also for fixed solutions. Another problem is that ultrasound monitoring employs liquid couplants, which makes it necessary to integrate a couplant dispenser into the UAV and a cleaning phase after the ultrasound measurement has been taken.

Corrosion is an extremely slow process, and corrosion rates between 0.1–0.2 mm/year are typical rates. Therefore, with a resolution of 1 μm, we will have to wait, on average, more than 4 days between consecutive measures to get statistically meaningful results. The work presented by References [32,33] proposes the use of curve fitting algorithms of a Gaussian echo model to estimate the time of arrival of ultrasound echoes. In order to improve the precision of the ToF measurements, it is necessary to compensate for temperature variations that affect both the thickness of the material and the speed of sound, digitally processing several back-wall echoes or using adaptive filtering techniques [34]. Other proposals rely on improving the Signal-to-Noise Ratio (SNR) and time resolution of the signals by employing the Wiener filter and spectral extrapolation [35]. An interesting idea to increase the SNR of the ultrasound system is to use coded excitation as in [36], which introduces a code gain in the signal's path. Finally, Ref. [37] proposes a non-standard approach based on cross-correlation, but, instead of performing the correlation with the transmitted signal, they correlate a back-wall echo with a reference signal that has been modified using an iterative algorithm that also involves cross-correlations. Thus, this approach achieves wall-thickness estimations (below 1 μm) on ultrasonic signals. However, this is at the expense of increased data processing complexity that will affect the power consumption and the cost of the solution.

To the best of our knowledge, none of the proposals in the literature develop a low-cost and low-power corrosion monitoring system based on ultrasound able to measure the thickness loss precisely showing that the solution can work for different thicknesses and mainly, for coated samples using the offshore standard coating NORSOK 7A. On top of that, the proposed approach aims to operate unattended inside offshore metallic towers.

### 3. Corrosion Monitoring System Concept

Our corrosion monitoring system design concept is focused on a novel and smart low-power and low-cost corrosion monitoring system for offshore wind turbines. The zones of interest are the splash zone and the atmospheric zone. The splash zone is the area that is intermittently exposed to seawater due to the action of tide or waves or both. This intermittence produces the highest level of corrosion. The atmospheric zone is the zone that is permanently exposed to marine air conditions. Therefore, the level of corrosion is also very high. Our approach includes two types of deployment, namely (i) a fixed and (ii) a mobile solution. In the case of the splash zone, the fixed sensors are attached to the structure due to the difficulties to access by a mobile platform. However, the atmospheric zone is covered by the mobile solution based on a drone flying inside the tower. Therefore, the main requirements to be achieved by our corrosion monitoring solution can be highlighted as follows.

- The monitoring solution shall be operational under the harsh conditions of offshore platforms.
- The corrosion sensor of the mobile and fixed solutions shall perform one measurement every month. However, this rate must be parameterizable with the aim of being increased when necessary.
- The corrosion sensor must be able to measure the ToF related to the thickness loss of the selected steel with a thickness between 40 mm and 70 mm but also with a thickness of 5 mm or 6 mm, which shall be the thickness of the produced samples to characterize the ultrasound solution.
- Be able to operate unattended for long periods of time.
- Both fixed and mobile solutions shall be able to perform corrosion measurements in the field and send those measurements wirelessly.
- The size of the overall final monitoring solution shall be up to $6 \times 11$ cm with a weight of up to 100 g. The sensor head shall be connected to the processing platform through wires.

### 4. Smart Ultrasound Sensor Design

Being aware of developing a reliable corrosion monitoring solution to decrease the cost of the maintenance and the downtimes during operation in the offshore wind sector, this paper presents a smart and robust corrosion monitoring system based on ultrasound technology and ToF technique that is capable of operating unattended inside an offshore metallic tower. With that aim, a prototype of the ultrasound test device has been designed. This is the Ultrasound Testbed (UT) with the capability of upgrading into a miniaturized solution that can be placed and operate inside of the wind turbine. UT is developed with the aim of performing frequent ultrasound measurements and analyzing the ultrasound responses. More details about sensor selection, excitation signals, and design methodology of the UT system are discussed in the following subsections.

*4.1. Probe Selection*

The selection of ultrasound probe or transducer needs to be done by considering the sensor characteristics, such as sensor dimensions, weight, dead zone duration, waveform duration, and peak frequency. The transducer operating frequency spectrum and the waveform duration are often provided by the manufacturer. The peak frequency of the transducer controls the penetrating power that affects the possible inspection thickness of test material, e.g., high-frequency ultrasound has low penetration power and might almost attenuate within the material. As at a fixed velocity in a perfectly elastic material the wavelength and frequency are inversely proportional, a change in the probe frequency will affect the sensitivity and the resolution to detect flaws in the material. Due to the diffraction of ultrasonic, the sensitivity is about half of the corresponding wavelength [38]. Thus, sensitivity and resolution are increased when probe frequency increases. The dead zone duration is the interval following the initial pulse where the transducer ringing

prevents detection or interpretation of reflected echoes. This is a constraint with thinner size samples as it is better that the time interval between back-to-back echoes has to be out of the dead zone.

As we are going to measure 5 mm thick samples, it is important to check the dead-zone of the selected probe. A 5 mm thick steel sample will introduce a delay between back-to-back echoes of $2 \times 5/5.9 \times 10^6 = 1.69\,\mu s$ (the speed of sound in steel is approximately $5.9 \times 10^6$ mm/s). This is the expected value of the ToF before corrosion takes place. This value is important when considering the dead zone of an ultrasonic probe. Dead zone value is highly dependent on the test conditions and instrumentation used. It is not normally given in the technical documentation of the probe and needs to be determined experimentally. The values shown on Table 1 correspond to the time when the ultrasound signal has almost completely settled (see Figure 2). However, it must be noted that the dead zone does not entirely invalidate the ultrasound data received on that interval. By proper filtering, and with the appropriate receiver's gain, it would be possible to extract echoes within the dead zone of the piezo.

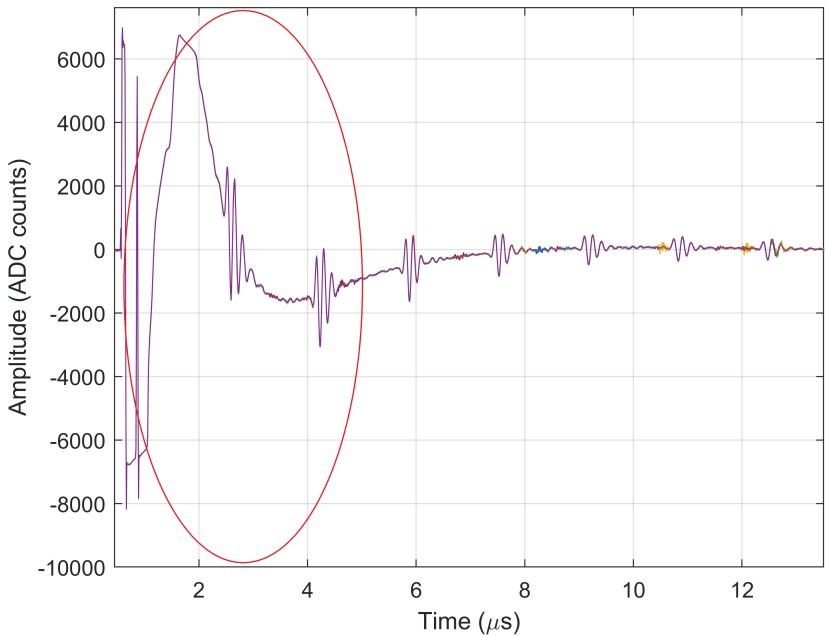

**Figure 2.** Dead Zone of an ultrasound probe.

**Table 1.** Performance and features of ultrasound probes.

| Part Name | YS15-10 | TSB10-06 | V111 |
|---|---|---|---|
| Manufacturer | YUSHI | OKOndt | Olympus |
| Diameter (mm) | 10 | 6 | 13 |
| Peak Frequency (MHz) | 7.86 | 10.2 | 8.44 |
| −6 dB Bandwidth (MHz) | 3 | 3 | 6.93 |
| Waveform Duration −20 dB Level (µs) | 0.7 | 0.840 | 0.272 |
| Dead Zone (µs) | 2.6 | 2.4 | 3.5 |
| Connector/Position | BNC/Top | Microdot/Side | Microdot/Side |
| Weight (g) | 50 | 7 | 20 |
| Price (€) | ~100 | ~100 | ~600 |

A comparative analysis of the performance specifications of three sensors has been done to select the most suitable one for the UT system as given in Table 2. As a result, V111 from Olympus [39] has been selected as the most suitable ultrasound probe for our system, specifically prioritizing its lower waveform duration (higher bandwidth) and sound pressure power. Having a higher dead zone compared to the other two sensors will not be a constraint in the real scenario as the system is designed for testing thickness around 40 mm.

**Table 2.** Applicability to the US system.

| Part Name | YS15-10 | TSB10-06 | V111 |
|---|---|---|---|
| Manufacturer | YUSHI | OKOndt | Olympus |
| Dead Zone | ✓✓ | ✓✓ | ✓ |
| Waveform Duration | ✓ | ✓ | ✓✓✓ |
| Bandwidth | ✓ | ✓ | ✓✓✓ |
| Sound Pressure Power | ✓✓ | ✓ | ✓✓✓ |
| Testing 5 mm Samples | ✓ | ✓ | ✓✓✓ |
| Testing 40 mm Samples | ✓✓ | ✓ | ✓✓✓ |
| Weight and Size | ✓ | ✓✓✓ | ✓✓ |
| Price | ✓✓✓ | ✓✓✓ | ✓ |

*4.2. Excitation Signal*

The conventional method is using a square wave as the excitation signal. The frequency response of the V111 peaks at 8.44 MHz; therefore, the frequency of the pulse must be similar in order to resonate and optimize the received signal's strength. Adding more pulses increases the signal's strength but at the expense of a larger pulse duration that deteriorates the time resolution of the ToF estimation. In our system, the period and the rise and fall times of the square wave can be controlled using a 125 MHz clock. The square wave pulse is bipolar ($\pm$15 V), and the generated frequency is obtained by dividing the clock frequency by an even integer $125/16 = 7.8$ MHz to guarantee a 50% duty cycle for positive and negative pulses.

*4.3. Signal Processing and Time-of-Flight Estimation*

Based on ultrasound waves, corrosion is evaluated by estimating the wall thickness loss that happens due to corrosion. The proposed approach is based on the Time of Flight (ToF) technique, which estimates the thickness of the test object by measuring the elapsed time between two consecutive echoes of the ultrasound response. This is based on the reflection of the ultrasound waves in different acoustic impedance conditions. This way, the effect of the couplant is greatly reduced because no changes in the couplant are expected in the time period of a ToF (in our case, less than 5 μs). The main drawback is the additional attenuation of the second echo that reduces the SNR of the signal and, therefore, the quality of the estimation. The determination of the time is done using the cross-correlation operation.

A block diagram of the signal processing operations performed to complete the ToF estimation in our approach is given in Figure 3. To have an accurate time base, it is important that both the ultrasound signal generator (US Generator) and the signal acquisition circuit (ADC Read) work with the same clock input. The filter removes unwanted frequencies components outside the bandwidth of the probe. The Echo Windowing block determines the location and size of the echo signals that will be cross-correlated. Some signal quality measurements are performed on the echo signals (signal level, signal width, echo amplitude and decay rate, separation between echoes, etc.). These measurements will help to detect low-quality signals and discard those ToF measurements in advance. Finally,

cross-correlation between two consecutive echoes, as well as peak detection and analysis, produce the ToF result.

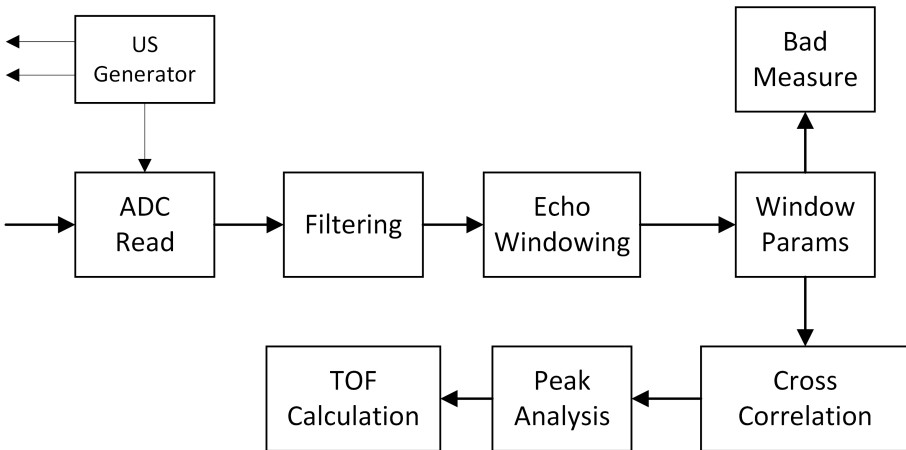

**Figure 3.** Block diagram of the signal processing chain.

*4.4. Design Methodology*

The design of a highly integrated monitoring system involves the collaboration between several design teams: analog circuits, digital signal processing, FPGA and microcontroller development, PCB prototyping and design, validation, etc. To coordinate the efforts from different teams, it is very important to have a system design methodology that takes you from the conceptual design to the final device to be deployed.

We have used a prototype-based design methodology in which the system is first designed from large building blocks that allow us to define the system using high-level abstraction languages, such as MATLAB or C/C++. As can be seen in Figure 4, the main blocks of the system prototype include a PC/Laptop and an Ultrasound Testbed. The latter is composed of a high-performance embedded processor that is able to generate and acquire ultrasound signals. The embedded processor runs a Linux system with the Debian distribution and includes an SD card for mass storage, Ethernet and Wi-Fi connections, and access to an internal FPGA that serves the purpose of buffering the ultrasound data from a high-speed 14-bit ADC. The Ultrasound Analog Front End (AFE) is composed of a pulser generator, adaptation circuitry, and a variable gain amplifier.

The system prototype permits us first to set up an experiment and gather raw ultrasound data for a variety of steel samples with different coatings, to test different piezo sensors, and change the experiment conditions, such as temperature, signal frequency, number of pulses, etc. The raw data is stored as a JSON file in a data repository that can be accessed from MATLAB to design and test signal processing and detection algorithms with the aim of estimating the time-of-flight from the ultrasound response.

Secondly, thanks to the embedded system, it is possible to run a C/C++ implementation of the algorithm in the Red Pitaya microprocessor and test it for quantization effects. In this case, the embedded system is responsible for estimating the time-of-flight, and the data obtained can also be uploaded to the data repository as a JSON file.

The hardware architecture of the final implementation is designed by analyzing the computing requirements of the algorithms designed in the prototype phase. We have chosen an architecture that includes a low-power Cortex-M4-based microcontroller (µC), an FPGA that interfaces with the µC using an SPI link and the AFE devices. The µC is in charge of supervising the system, i.e., powering on and off the different devices (FPGA, AFE, memories), launching measurements and gathering the results and raw data from the FPGA using the SPI interface and providing communication interfaces with the external world (UART and wireless communications). The high-speed sampling of the ultrasound signals has called for the use of an FPGA. Ultrasound signal generation is also handled by

the FPGA to produce synchronized signals for the pulser. The processing of ultrasound data requires intensive computation that is more efficiently performed in the FPGA.

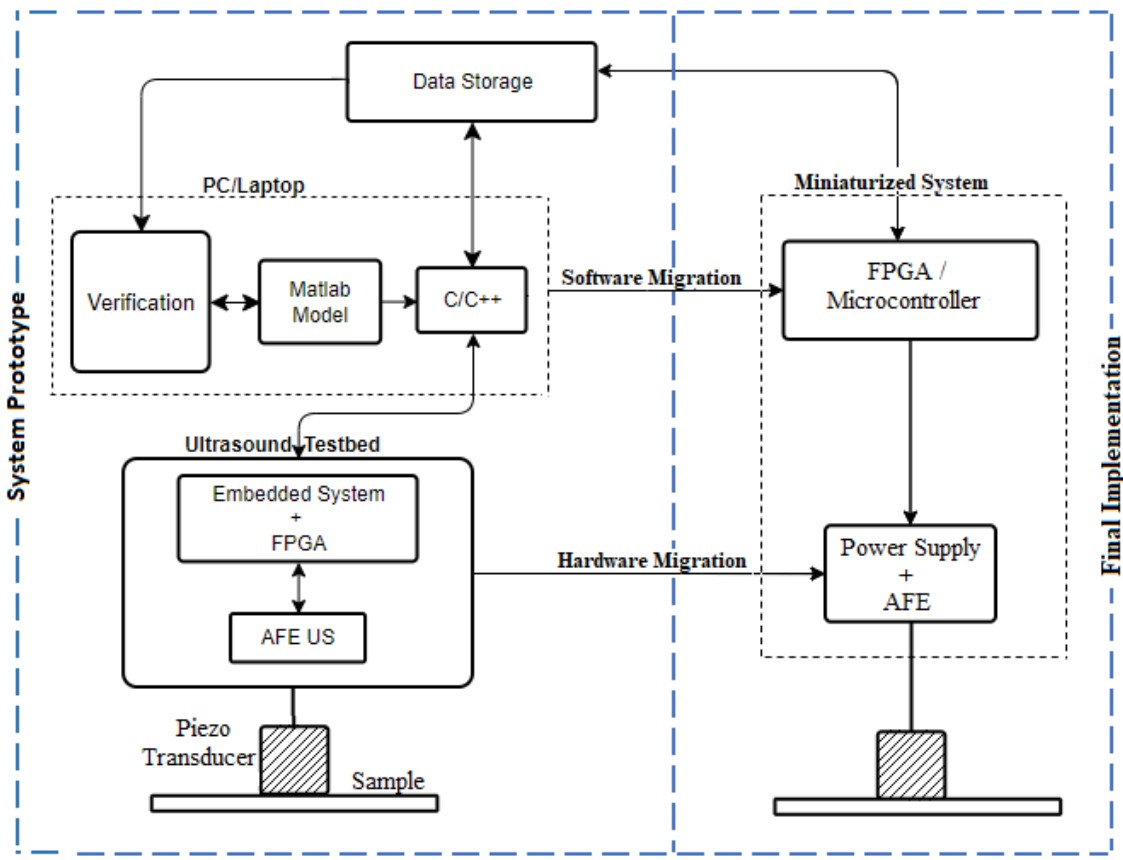

**Figure 4.** System design methodology.

By having models of the processing of ultrasound signals at different abstraction levels (MATLAB, C/C++, and FPGA), it is possible to perform validations at different stages of the design process as it advances towards the final implementation using the raw data stored in the data repository (see Figure 4). In this way, we can feed the MATLAB model with the real data from the miniaturized system and compare the ToF results obtained for verification.

Finally, after validating the ultrasound test method and algorithm implemented in UT, it has been migrated to a miniaturized system, as shown in Figure 5 (right), being able to place it inside a wind turbine to conduct ultrasound tests in the real scenario. The performance specifications of the miniaturized solution are given in Table 3.

**Table 3.** Miniaturized ultrasound system performance.

| Specification | Value |
|---|---|
| Dimensions | $(110 \times 60 \times 15)$ mm |
| Power supply | Battery 3.6 V, 5800 mAh |
| Average power consumption (Sleep mode) | 5.4 µW |
| Average power consumption (per measurement event) | 850 mW |
| Wake Up Time before US measurement | 1 s |
| Measurement Time | 200 µs |

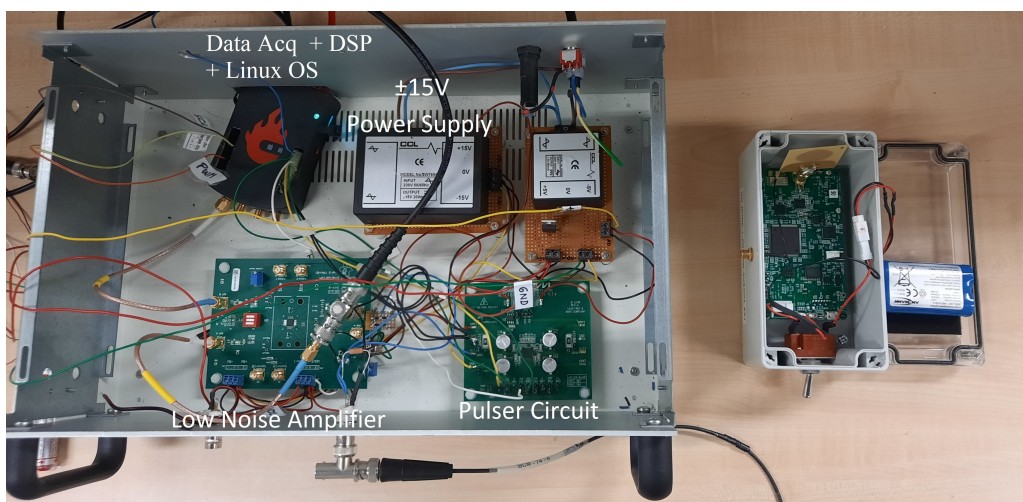

**Figure 5.** Pictures of the UT (**left**) and miniaturized solution (**right**).

## 5. Sample Preparation and Experimental Setup

### 5.1. Sample Preparation

Two types of S355 structural steel samples have been used for the system performance validation as reference samples and test samples. Reference samples have been used mainly to verify the test method and evaluate the precision of the measurements during the algorithm development. On the other hand, test samples have been used to test the long-term system performance. The dimensions of the reference and test samples are shown in Table 4.

**Table 4.** Reference and test sample dimensions.

| Sample Size | (75 × 150 × 5) mm | (75 × 150 × 40) mm |
| --- | :---: | :---: |
| Reference Samples | ✓ | ✓ |
| Test Samples | ✓ | – |

The real thickness of a wind turbine tower is around 40 mm. However, the test samples have been used only with 5 mm thickness due to the practical difficulties in transportation and handling of the samples during the frequent experiments.

Coating layers are applied to protect the substrate from reacting with its environment so that it will not engage in an electrochemical process and corrode. Coatings provide protection against moisture, dissolved gases, acids, and other reactants in the environment. The test coated samples with applied standard coating NORSOK 7A (see Figure 6) were tested by our ultrasound system to observe the changes in the characteristics of the ultrasound signal and to analyze the performance of the developed signal processing algorithm.

Corrosion is a very slow process, and it takes a long time to induce corrosion on coated samples. Therefore, in order to observe a measurable thickness loss with time, the corrosion process has been accelerated by producing a 2 mm-wide and 50 mm-long scribe, cut completely through the coating, which is made in the coated samples, as can be seen in Figure 6. This scribe emulates in some way a coating defect or failure. The prepared coated samples were exposed and conditioned based on cyclic aging resistance processes (cyclic processes of exposure to UV, neutral salt spray, and low-temperature exposure) in agreement with EN ISO 12944-9. The test method consists of 25 cycles of 4200 h of exposure: over three days, the samples are exposed to UV/condensation according to EN ISO 16474-3; during the following three days, the samples are exposed to neutral salt spray according to EN ISO 9227; during the next day, the samples are exposed to low temperature at ($-20 \pm 2$ °C).

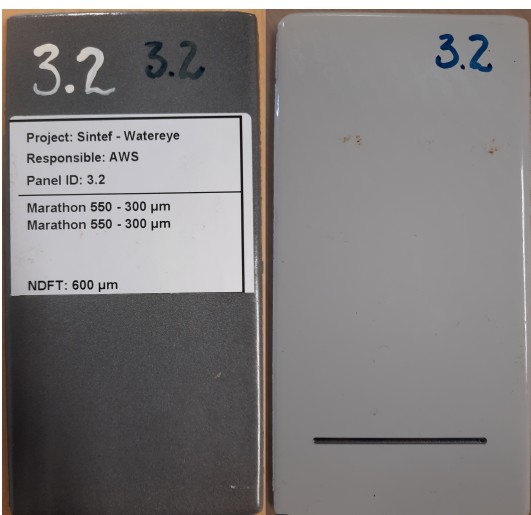

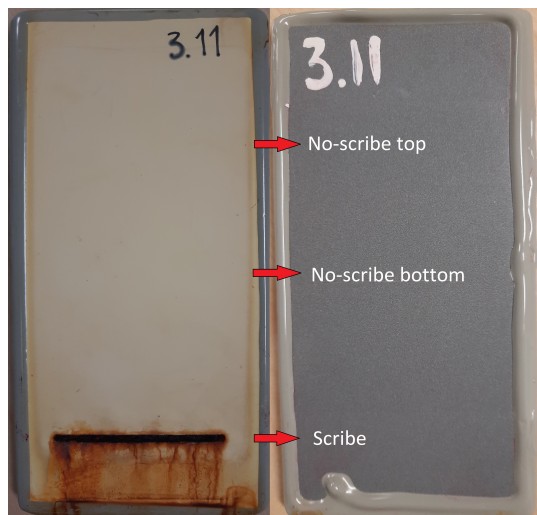

**Figure 6.** Picture of coated test samples: non-exposed coated test sample (**top**) and three months exposed sample (**bottom**).

In order to observe and measure the corrosion process, the coated samples have been exposed to cyclic aging resistance process for different time durations as 1 month, 3 months (see Figure 6 (bottom)), and 6 months. Testings of coated samples have been performed in three different layers as: on scribe, no-scribe bottom layer, and no-scribe top layer, to get an indication of thickness loss due to corrosion under cyclic aging process.

### 5.2. Experimental Setup

The developed Ultrasound Testbed (UT) prototype is shown in Figure 5 (left). It consists of a low-noise amplifier, pulser, piezoelectric transducer, and FPGA controller with high-frequency ADC and DAC conversion. The UT system is capable of performing frequent ultrasound testings. In the purpose of performing tests and analyzing results, the UT is connected to a PC/Laptop with MATLAB via Wi-Fi or Ethernet, and the experiments are launched and controlled via PC/Laptop. The main advantages of the UT system are that it allows the user to perform frequent tests, to visualize and compare the results easily, and to develop, analyze, and update the signal processing algorithms for better ToF estimations.

The ultrasound test is performed by placing the sensor on the sample using a couplant applied as shown in Figure 7. Olympus D12-gel, that is recommended for rough surfaces, has been used as the couplant to perform the ultrasound tests. Each test is performed by launching 25 excitation signals in a row and collecting the corresponding response from the test sample. Therefore, the ToF value calculated for a particular location is an average

of 25 measures, and this helps to minimize the uncertainty of the results. In addition to that, the precision is calculated as the standard deviation of those 25 measurements.

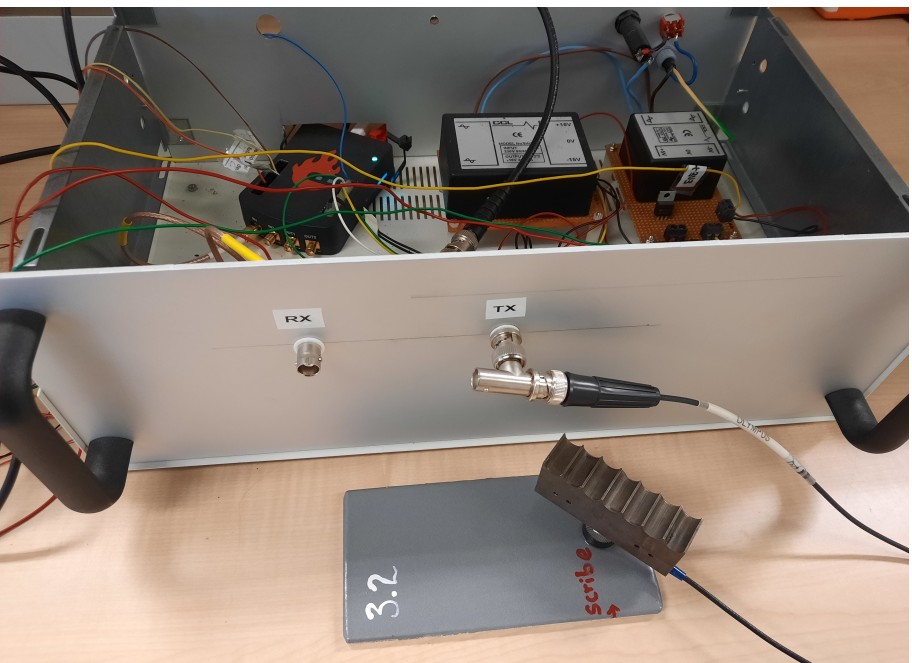

**Figure 7.** Non-exposed coated sample: on scribe measurements.

Ultrasonic couplant is used to make good contact between transducer and test piece, facilitating quality sound energy transmission. In NDT, the couplant is used because they do not transmit effectively through air. Even so, the applied couplant layer still can affect to ultrasonic velocity and attenuation of the back wall echoes from the test piece. This depends on the acoustic impedance of the couplant being used and how good the contact is made between the test surface and the transducer. On the other hand, if the amount of couplant is excessive, a couplant layer can act as a wedge and alter the direction of the sound wave affecting the final result.

The measurement of ToF in ultrasound signals propagating through steel is directly related to the variation of the speed of sound with temperature and, to a lesser extent, to thermal expansion. Therefore, we need to measure the temperature of the steel to compensate for the temperature effect in the measurement of ToF. However, all the measures performed using UT were made at room temperature. Therefore, no temperature compensation is needed at this point.

The value of Time of Flight depends on the speed of the sound through the target material. The speed of sound in steel is approximately 5900 m/s [40]. To increase the accuracy of the measurements, the speed of sound has been calculated for the selected steel material (S355J2G3) using the 5 mm reference samples. In the process of calculating the speed of sound, the reference bare steel sample surface was divided into 8 zones, and the average thickness of each zone was measured using a digital micrometer. Considering it as the thickness reference, the average speed of sound for each zone was calculated using Equation (1), based on 5 ToF measures performed by using the UT on each zone. Accordingly, the final calculated average speed of sound at room temperature for bare steel reference samples is 5950 m/s. This value has been set as the speed of sound in steel for the rest of the test samples measured by the ultrasound solution.

$$\text{Speed of sound [m/s]} = \frac{2 \times 10^3 \times \text{Reference thickness of sample [mm]}}{\text{Time of Flight (ToF) [\mu s]}}. \tag{1}$$

The UT test approach has been validated by comparing the results from UT and micrometer measures from reference samples of both 5 mm and 40 mm thickness. The results showed a good agreement between the thickness estimations done by the UT and by conventional methods, such as using a caliper or a micrometer.

## 6. Results and Discussion

### 6.1. Test Samples Measurements

Though the test method of UT system has been validated using reference samples, the long-term system performance has been analyzed by measuring the test samples, as explained in Section 5.1. The following subsections discuss the results obtained by the UT system for non-exposed and exposed coated test samples.

### 6.1.1. Measurements Precision of Non-Exposed Test Samples

The time response of an ultrasound sensor collected from a test object shows the characteristic behavior of echoes bouncing back and forth the wall's boundaries.

As discussed in Section 4.3, after ADC reading follows a filtering process that includes a bandpass filter that is used to remove the unwanted frequency components from the signal. Figure 8 presents the bandpass filtering results obtained for non-exposed bare steel and coated samples. Figure 8 (top) shows the ultrasound response from the bandpass filter for a bare steel sample that has a well separated and small echo duration, which facilitates the separation of the consecutive echoes for the ToF calculation. In addition, compared with Figure 8 (bottom), it can be seen that the echo duration is longer for coated samples compared to the bare steel samples. This fact makes the echo separation more difficult in the case of coated samples. The estimated echo durations for bare steel and coated samples are around 0.7–0.9 μs and 1.3–1.5 μs, respectively.

The precision of the ultrasound measures gives important information about the system's performance. Having a good precision allows us to take more frequent corrosion measurements, which can potentially be used to estimate the corrosion rate at any instant by using a derivative FIR filter. The latency of the corrosion rate measurements using this method will be in the range of a few hours which, in practical terms, can be considered a real-time signal for an O&M operator. Figure 9 shows the data plotted between the obtained ToF precision ($\sigma_{ToF}$) and the signal level (SL) for both non-exposed bare steel and coated samples. Here, the SL refers to the Root Mean Square (RMS) value of the first detected echo measured after the amplifier. This value is calculated digitally after the bandpass filter mentioned above. Because of this, SL is an indication of the SNR of the digitized signal after bandpass filtering.

The precision of ToF values we are plotting here is the standard deviation of 25 measures launched at one location, as discussed in Section 5.2. The data points of the graph for non-exposed coated samples in Figure 9 are related to both on scribe and no-scribe areas. It has been clearly observed, in the Figure 9, that the SL of ultrasound measures has considerable variations, depending on the position of the piezo, and, when the signal level decreases, the precision obtained for those respective measures gets worse.

The data points of signal level versus ToF precision were best fitted with an exponential decay function ($\sigma_{ToF} = A \cdot e^{-SL/B}$). Each fitted curve is represented over the signal level range of 35 mV to 250 mV. Considering the decay constant coefficient associated with each fitted curve (parameter $B$ in mV indicates the increase in SL necessary for a 63% reduction in $\sigma_{ToF}$), the ToF precision versus SL behavior is almost the same for the bare steel and coated samples. Both bare and coated samples have good ToF precision results, but the obtained SLs are slightly better in coated samples. The reason behind this behavior may be that the quality of contact made between the transducer and the coated samples was better than in the case of the bare steel samples. Furthermore, the results do not show any significant reduction of the signal level due to the coating layer.

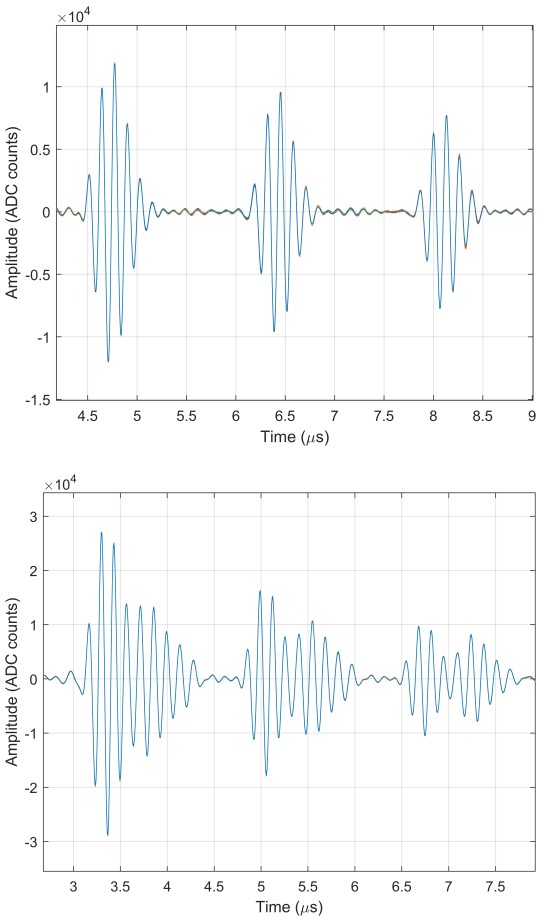

**Figure 8.** Bandpass filter results for bare steel sample (**top**) and coated sample (**bottom**).

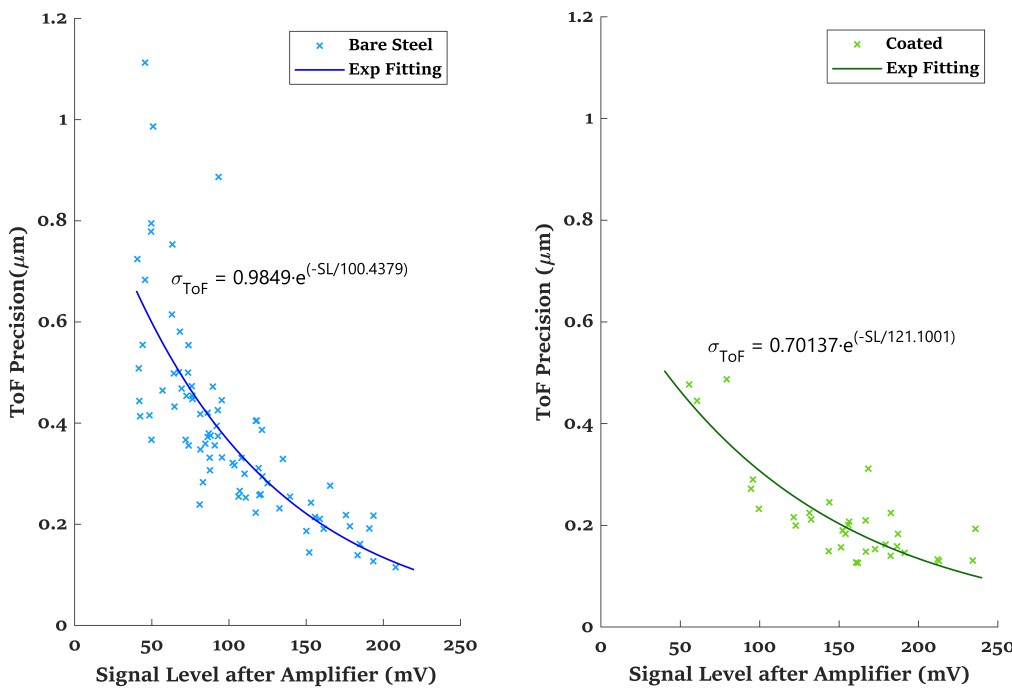

**Figure 9.** Non-exposed samples: signal level versus precision.

6.1.2. Measurements Precision of Exposed Test Coated Samples

Different sets of numbered coated samples have been exposed to a controlled corrosive environment under the cyclic aging test after 1 month, 3 months (see Figure 6 (bottom)), and 6 months exposure. After the completion of the exposure time, each sample has been tested using the UT. The ultrasound thickness measures have been performed in both on scribe and no-scribe locations of the coated samples, as discussed in Section 5.1.

The precision of measures obtained for the test coated samples exposed under cyclic aging test for different time durations with respect to the signal level has been plotted in Figure 10. Here, also, the SL refers to the Root Mean Square (RMS) value of the first detected echo measured after the amplifier, and precision refers to the standard deviation of 25 measures performed in the same location ($\sigma_{ToF}$). Comparing the precision of ToF measures between on scribe and no-scribe areas, it can be seen that, in order to obtain similar precisions in Scribe and Non-Scribe areas, we must increase the SL by ~75 mV (see the *B* parameter of fitting curve) when measuring Scribe areas.

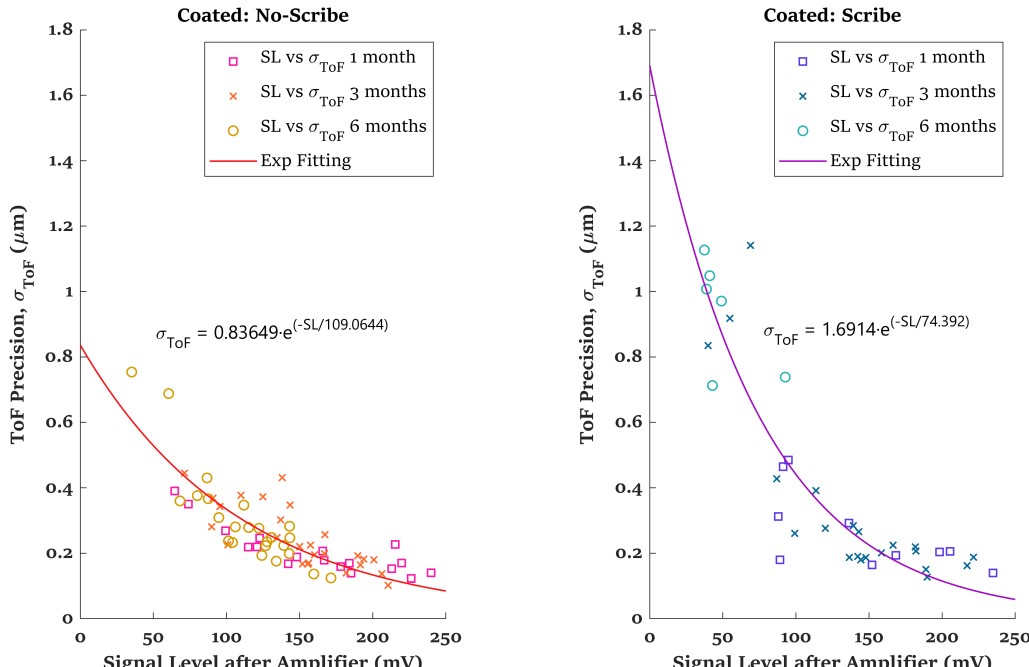

**Figure 10.** Exposed coated samples: signal level versus precision.

Furthermore, in the case of the on scribe area, it can be observed that the most lower signal level data points belong to the samples with higher exposition time. This may be caused due to the non-uniformity of the corrosion layer developed on the scribe area. In general, we are obtaining a good precision for both bare and coated samples. The ToF precision computed here is the standard deviation of ToF values of 25 measures at the same sensor position (as discussed in Section 5.2). If we place the sensor on different locations of the same sample and calculate the standard deviation of ToF measures, the obtained precision is lower than the given values here. This is because the thickness at different locations of the same sample varies, even for the non-exposed bare steel samples. These small thickness variations (at µm level) are due to the common existing imperfections during the sample production. Therefore, in order to have a good precision of ToF measures, the positioning of the sensor on the same location is very important. As discussed in Section 3, from the two modes of operation (fixed and mobile), the positioning accuracy can be a constraint for the mobile solution, which we have to address in future work.

6.1.3. Thickness Loss Measurements on Exposed Test Coated Samples

Each exposed coated sample has been measured using UT, as mentioned in Section 5.1. Figure 11 shows the thickness measures on scribe and no-scribe locations of coated samples

for different exposed time durations. Considering the fact that no significant difference has been observed between the no-scribe bottom and no-scribe top layer thickness measures, only an average of no-scribe top and bottom layer measures has been used as the no-scribe measures.

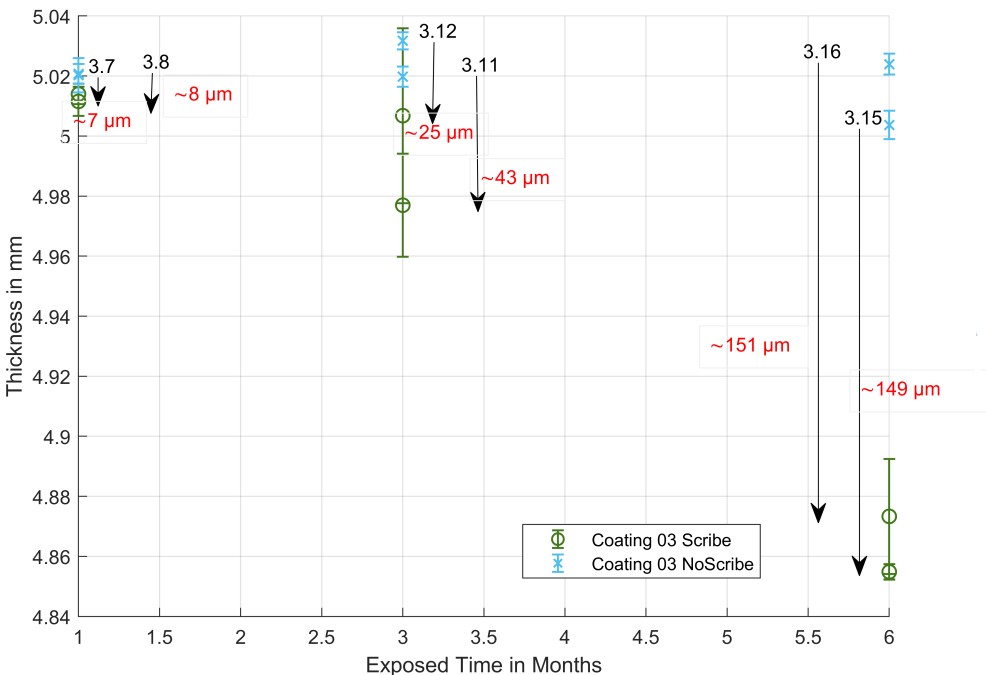

**Figure 11.** Coated samples: thickness versus exposed time in months.

The detected thickness differences between the scribe and no-scribe locations of the same sample are indicated by the arrows. With the scribe made in the coated samples, the corrosion process has been accelerated and initiated along with the scribe. Apparently, the results show that the thickness losses are higher on the scribe area than on the no-scribe area when the exposure time is sufficient to produce a significant thickness loss due to corrosion, as is the case of 6 months exposure.

## 7. Conclusions

In this paper, we present a corrosion monitoring system based on ultrasound technology. The size, weight, and other performance parameters, such as precision in thickness loss measurements, power consumption, and wireless connectivity, make the proposed solution very suitable for unattended deployment in offshore wind turbines or installed in mobile platforms, such as drones operating inside the tower, to cover easily large structures. This mobile solution would be able to detect earlier critical failures due to corrosion and, consequently, plan better maintenance actions.

The system design was developed using an Ultrasound Testbed (UT) that allows us to assess the performance of the detection and measuring algorithms in MATLAB and C/C++. The UT can launch ultrasound experiments with different parameters (frequency, waveform duration, gain, etc.) in order to find the optimal values. As a result, we created a large database of ultrasound raw signals that were later used to validate the algorithms implemented in the final miniaturized system. After validation, the UT system has been successfully migrated into a miniaturized system with low power, low cost, and size of $110 \times 60 \times 15$ mm.

All the measurements presented in the paper were done using the UT and analyzed under a MATLAB development environment for test samples of $75 \times 150 \times 5$ mm. However, the system was validated for reference samples of $75 \times 150 \times 40$ mm, as well, demonstrating that the solution is flexible enough to work with different thicknesses closer to a

realistic scenario. Further large-scale validation of the proposed solution mainly for sensor placement and alignment in the case of the mobile solution will be done in the near future. Related to that, and taking into account the harsh conditions of these offshore platforms, we pay special attention to the batteries and sensor probe with the aim of reducing the number of replacements and to facilitate those replacements when needed.

Considering the results obtained in this research work, the signal level (SL) and ToF precision measurements show a similar exponential decaying relation for both bare steel and coated samples. In general, we could observe that the signal level is slightly higher for coated samples than for bare samples and that precisions below 1 μm can readily be obtained for SL > 50 mV. The thickness loss could be estimated in the coated (NORSOK 7A system) samples as the difference in the sample thickness between the no-scribe and on scribe areas after running the cycling aging test for several months. The results show, as expected, a growing thickness loss as exposition time increases. The values of the thickness loss reported is an average of the difference between the thickness along the on scribe area and the thickness in many no-scribe areas of the same sample. Other observations are that SL in on scribe areas is normally worse than in no-scribe areas, SL deteriorates as exposition time increases, and ToF precisions are again below 1 μm for SL > 50 mV.

Therefore, the thickness estimation of coated samples exposed under the cyclic aging test shows meaningful information about thickness loss due to corrosion. The variable tolerance of the wall thickness at different locations of the same sample very well obliges sensor positioning for the mobile platform. However, this is not an issue for the fixed solution.

To sum up, it can be said that the main objectives related to achieve a miniaturized system, to get a precision of 1 μm, and to estimate the thickness loss with high precision have been accomplished, taking into account bare and coated samples. The obtained precision allows us to take several thickness measurements per day to estimate the value of the corrosion rate in real-time for practical purposes.

**Author Contributions:** Conceptualization, A.C. and A.I.; Formal analysis, A.I. and U.C.T.; Funding acquisition, A.C. and A.I.; Investigation, A.C., A.I. and U.C.T.; Methodology, A.I. and U.C.T.; Project administration, A.C.; Software, A.I. and U.C.T.; Supervision, A.C. and A.I.; Validation, A.I. and U.C.T.; Writing—Original draft, U.C.T.; Writing—Review & editing, A.C. and A.I. All authors have read and agreed to the published version of the manuscript.

**Funding:** This work was supported by the WATEREYE project, which has received funding from the European Union's Horizon 2020 research and innovation program under grant agreement No. 851207.

**Institutional Review Board Statement:** Not applicable.

**Informed Consent Statement:** Not applicable.

**Data Availability Statement:** Not applicable.

**Acknowledgments:** This work has been possible thanks to the cooperation of CEIT with all the WATEREYE partners, especially in this case with SINTEF Industry, who is responsible for producing the samples, measuring the samples through conventional methods, and corroding the samples in lab.

**Conflicts of Interest:** The authors declare no conflict of interest.

## Abbreviations

The following abbreviations are used in this manuscript:

| AFE | Analog Front End |
|---|---|
| CAPEX | Capital Expenditure |
| GWEC | Global Wind Energy Council |
| NDT | Non-Destructive Testing |
| O&M | Operation and Maintenance |
| OWT | Offshore Wind Turbines |
| PCB | Printed Circuit Board |
| ROI | Return on Investment |
| SHMS | Structural Health Monitoring Systems |
| SNR | Signal-to-Noise Ratio |
| WSN | Wireless Sensor Network |
| SL | Signal Level |
| ToF | Time-of-Flight |
| UAV | Unmanned Aerial Vehicle |
| UT | Ultrasound Testbed |
| WT | Wind Turbine |

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
