# Peer review of "Ultrasound-Based Smart Corrosion Monitoring System for Offshore Wind Turbines"

_applsci, doi:10.3390/app12020808_

Round 1
Reviewer 1 Report
In the manuscript titled “Ultrasound based Smart Corrosion Monitoring System for Offshore Wind Turbines”
The authors have submitted their findings on smart corrosion monitoring using ultrasound pulse-echo technique to predict corrosion of offshore wind turbines structures in the atmospheric and splash zone. The corrosion prediction is based on thickness measurement of bare steel and coated scribed and unscribed samples using miniatured UT testbed for different time duration and in different environment (UV, salt spray and low temperature).The results from the experiment demonstrate good precision of predicting corrosion based on ToF and signal level measurements for both coated and bare steel samples. This manuscript shows UV pulse-echo as a promising technology that can be used to predict corrosion in offshore wind turbines with good precision.
The authors can improve this manuscript if they can add these below items if possible, to make it more comprehensive-
- I observe that the authors have not discussed in detail the methods for UV, neutral salt and low temperature. Giving more info in the method section will make this manuscript more comprehensive and accurate.
- Provide qualitative microscopic images of exposed samples would add more detail to the manuscript,
- Provide info on converting thickness measurement into corrosion rates ( mpy).
Reviewer 2 Report
In this paper, an ultrasound-based NDT for continuous corrosion monitoring for offshore wind turbines is discussed. It includes the design stage, the measurements precision and verification of the system.
The subject of the paper is interesting and offers one of the possible solutions for fast non-destructive corrosion monitoring in structures with limited access, like off-shore wind turbines. The level of English is good, and the text is fluently legible.
However, the structure of the paper is a bit confusing, more like a report. There are no clear boundaries between the introduction / state-of-the-art part and chapters describing the design stage of the developed system. Therefore introduction part of the paper seems rather extensive and detailed compared to the experimental and results part. It is very difficult to distinguish between the introduction part, development part, verification part (experimental & results) and conclusions. I would recommend making this clearer.
Some additional comments should be considered before the paper is published:
- There are no references in subchapter 1.2, although it is (also) summarizing the existing solutions of corrosion inspection and monitoring.
- The splash zone was well defined as the most extreme in the marine (off-shore) environment. Are there any other critical parts on the micro level, e.g. crevice, joints, galvanic couplings, welds etc. where corrosion activities could be more intensive? Are there any limitations for the use of developed NDT on these parts?
- 200 micro-m/year does not seem very slow corrosion rate. What is the critical thickness loss of the steel wall? In the presence of chlorides and defects on coating also the local corrosion of even higher rates should be expected. How is developed NDT distinguishing between local and general corrosion damages?
- Validation was made only on the small-scale samples. The consideration about using this NDT on a larger scale should be done in conclusions, e.g. advantages, disadvantages and open questions.
Publication of this paper requires minor changes.
Reviewer 3 Report
This paper is well organized and it can be accepted as it is.Author Response
There are no comments to provide a response.
Reviewer 4 Report
Offshore wind turbine are widely applied in wind energy industry. However, Corrosion is one of the main root for offshore structural failure. Authors studied the ultrasound pulse-echo technique used in corrosion monitoring system. Some results have been carried out. However, the current form of this study cannot be acceptable. Some aspects as listed below:
- Ultrasound technique is widely used in corrosion monitoring. More details about ultrasound pulse-echo technique should be given. What are the advantages over traditional in corrosion monitoring?
- In Fig. 5, How to ensure the accuracy of the system?
- More discussion about the smart corrosion monitoring system should be given. What is the smart part?
- The image quality in Fig 10-12 should be improved.
- The content is partial to engineering, and lack of innovation.
Round 2
Reviewer 4 Report
Offshore wind turbine are widely applied in wind energy industry. However, Corrosion is one of the main root for offshore structural failure. Authors revised the manuscript according to the commands. However, the current form of this study cannot be acceptable. Some aspects as listed below:
- In Fig.9 and Fig,10, the The meaning of the symbol σToF should be given firstly. How to use the exponential decay function of signal level vs ToF precision?
